# Efficient preservation of sprouting vegetables under simulated microgravity conditions

Yoshio Makino[1]*, Kanji Ichinose[1], Masatoshi Yoshimura[1], Yumi Kawahara[2], Louis Yuge[2,3]

1 Graduate School of Agricultural and Life Sciences, The University of Tokyo, Tokyo, Japan, 2 Space Bio-Laboratories Co., Ltd., Hiroshima, Japan, 3 Graduate School of Biomedical and Health Sciences, Hiroshima University, Hiroshima, Japan

* amakino@mail.ecc.u-tokyo.ac.jp

**Data Availability Statement:** All relevant data are within the manuscript.

## Abstract

The effectiveness of a simulated microgravity environment as a novel method for preserving the freshness of vegetables was investigated. Three types of vegetables were selected: vegetable soybean, mung bean sprouts, and white radish sprouts. These selected vegetables were fixed on a three-dimensional rotary gravity controller, rotated slowly. The selected vegetables were stored at 25°C and 66% of relative humidity for 9, 6, or 5 d while undergoing this process. The simulated microgravity was controlled utilizing a gravity controller around 0 m s$^{-2}$. The mung bean sprouts stored for 6 d under simulated microgravity conditions maintained higher thickness levels than the vegetable samples stored under normal gravity conditions (9.8 m s$^{-2}$) for the same duration. The mass of all three items decreased with time without regard to the gravity environment, though the samples stored within the simulated microgravity environment displayed significant mass retention on and after 3 d for mung bean sprout samples and 1 d for white radish sprout samples. In contrast, the mass retention effect was not observed in the vegetable soybean samples. Hence, it was confirmed that the mass retention effect of microgravity was limited to sprout vegetables. As a result of analysis harnessing a mathematical model, assuming that the majority of the mass loss is due to moisture loss, a significant difference in mass reduction coefficient occurs among mung bean sprouts and white radish sprouts due to the microgravity environment, and the mass retention effect of simulated microgravity is quantitatively evaluated utilizing mathematical models. Simulated microgravity, which varies significantly from conventional refrigeration, ethylene control, and modified atmosphere, was demonstrated effective as a novel method for preserving and maintaining the freshness of sprout vegetables. This founding will support long-term space flight missions by prolonging shelf life of sprout vegetables.

## Introduction

Vegetables constitute an important food staple, forming a source of micronutrients containing vitamins and minerals [1]. However, it is quite difficult to store fresh vegetables and maintain their freshness after they have been harvested, and the loss/waste rate of vegetables during the

**Funding:** Space Bio-laboratories Co., Ltd. (SBL) provided support in the form of salaries for author [Y. K.], but did not have any additional role in the study design, data collection and analysis, decision to publish, or preparation of the manuscript. The specific roles of these authors are articulated in the 'author contributions' section.

**Competing interests:** We have the following interests: Yumi Kawahara is the president of Space Bio-laboratories Co., Ltd. There are no patents, products in development or marketed products to declare. This does not alter our adherence to all the PLOS ONE policies on sharing data and materials, as detailed online in the guide for authors.

distribution process is 18% in developed countries and 46% in developing nations [2]. The causes of vegetable loss are due to wilting, loss of L-ascorbic acid [3], discoloration [4], and spoilage [5]. Several methods for preserving vegetable freshness have been studied and analyzed.

Among these methods investigated, refrigeration is the most versatile and prevalent freshness preservation method for vegetables. Li et al. [6] reported the effect of retaining visual acceptability among refrigerated produce, and suppressing mass loss and decay for eight types of items stored at 6°C. Controlled atmosphere (CA) storage, which suppresses respiration under low $O_2$ and high $CO_2$ environments, demonstrates a significant freshness-keeping effect when used in combination with refrigeration, and is employed for the long-term storage of apples [7]. A storage unit is harnessed for CA storage. In contrast, modified atmosphere packaging (MAP), which is obtained by achieving the same effect using a plastic pouch, stops the spoilage of tomatoes [8], the softening of persimmons [9], and suppresses the yellowing of soft kale [10] and vegetable soybeans [11]. The inactivation of ethylene gas as an aging hormone is an effective freshness-keeping method, and a high number of freshness- effects derived from the ethylene-inactivating regent "1-methylcyclopropene" were reported as introduced in a review by Watkins [12]. Heat treatment has been identified as possessing the effect of sterilizing microorganisms, deactivating enzymes, and inducing heat shock protein to maintain freshness in vegetables [13]. Akbudak et al. [14] reported that heating vegetables at 54°C for 5 min was effective for preserving the freshness of tomatoes. However, currently, an impactful freshness preservation method has not been employed practically other than the aforementioned freshness maintenance technologies. Therefore, a novel method of maintaining freshness in vegetables is highly desirable to improve the produce marketplace and industry's storage abilities.

The wilting of vegetables due to mass loss is an important cause of degradation in vegetables after the harvest period. The majority of the mass loss in vegetables after harvest is due to moisture loss, which is mainly associated with transpiration [15]. Hirano and Kitaya [16] reported that transpiration increases with higher levels of gravity over the course of a parabolic airplane flight experiment utilizing grown strawberries as a plant sample. This phenomenon of increased transpiration may also be observed within postharvest vegetables. Thus, it is hypothesized that the transpiration rates and mass loss of vegetables postharvest can be reduced by placing the vegetables under a controlled microgravity environment.

In recent years, a device capable for creating and maintaining a microgravity environment on Earth has been developed and harnessed to investigate biological reactions in outer space. The principle of operation of the microgravity device is to rotate the object 360° slowly in order to bring the mean value of gravity close to 0 m s$^{-2}$. To be precise, the created conditions are different from the microgravity conditions found in outer space, so the machine is suitably referred to as a simulated microgravity device. Several studies published investigating the applications of this device include the homogenization of crystals [17], the attenuation of rat myogenic differentiation [18], and the effects of microgravity on epidermal stem cell metabolism [19]. Experiments utilizing plants have reported microRNAs in tomato that respond to gravity [20] and the delay of rice flowering in a simulated microgravity environment [21]. However, no report has been found concerning changes in the quality of horticultural products after harvest in relation to an experimental microgravity environment.

Therefore, the objective of this study was to demonstrate the effectiveness of vegetable storage and freshness preservation facilitated by a simulated microgravity environment as a new method for produce quality protection postharvest. Since a stable supply of food is important for long-term stays in space, research on food production in space environment has been

conducted in recent years [22]. The results of this study will support long-term stay missions by extending the shelf life of the crop.

## Materials and methods

### Samples

After a single day following harvest, vegetable soybeans (*Glycine max* (L.) Merr.), mung bean sprouts (*Vigna radiata* (L.) R. Wilczek), and white radish sprouts (*Raphanus sativus* var. *longipinnatus*) were purchased from a supermarket in Tokyo.

### Storage method

Vegetables were prepared in an A-04 Fix Box (AS ONE Corp., Osaka) (Fig 1(A)). Five sheaths of vegetable soybeans, five mung bean sprouts, or three bundles of white radish sprouts (four sprouts per bundle) were fixed within each box. Four boxes were prepared for each type of vegetable and then divided into two groups. Since the box has a vent, the samples were kept under normoxia ($O_2$ 21%, $CO_2$ 0.04%).

One group of vegetables was attached to the inner flame of the gravity controller (Gravite®, Space Bio-Laboratories Co., Ltd. Hiroshima, Japan) affixed with curing tape as shown in Fig 1 (B) and stored under simulated microgravity conditions by rotating the inner and outer flames 360˚ at 4 and 2 rotations per minute, respectively. The controller was set in a thermohygrostat (25˚C, relative humidity 66%). Another study group was stored in the same thermohygrostat as the control sample. Storage periods were 9, 6, and 5 d for vegetable soybeans, mung bean sprouts, and white radish sprouts, respectively.

### Shooting of mung bean sprouts using computer vision system

Mung beans sprouts were shot on 0 and 6 d to identify the form of the samples using the same computer vision system (FMVU-13S2C-CS, Point Grey Research Inc., Richmond, British Columbia, Canada) as the previous report [23]. A clear outline of a sample was traced utilizing the outline detection filter of Adobe Photoshop ver. 13.0.1 (Adobe Systems Inc., San Jose, CA).

### Measurement of mass retention rate

The mass retention rate was calculated from the measured mass value following this equation [11]:

$$M_r = 100 \bullet M_t/M_0 \tag{1}$$

where $M$ denotes the mass of a sample (g), subscript $r$ denotes the retention rate, and $t$ denotes the storage period, with zero denoting the initial day.

### Analysis of mass loss velocity

The majority of mass loss in vegetables postharvest is caused by moisture loss [15]. Therefore, the moisture diffusion model was coopted for analyzing the mass reduction velocity of the vegetables in the present study. In addition, the influence of gravity on the mass reduction rate of sampled vegetables was mathematically analyzed using the following equation [24, 25] on the basis of moisture diffusion:

$$(M_t - M_e)/(M_0 - M_e) = \exp(k \bullet t) \tag{2}$$

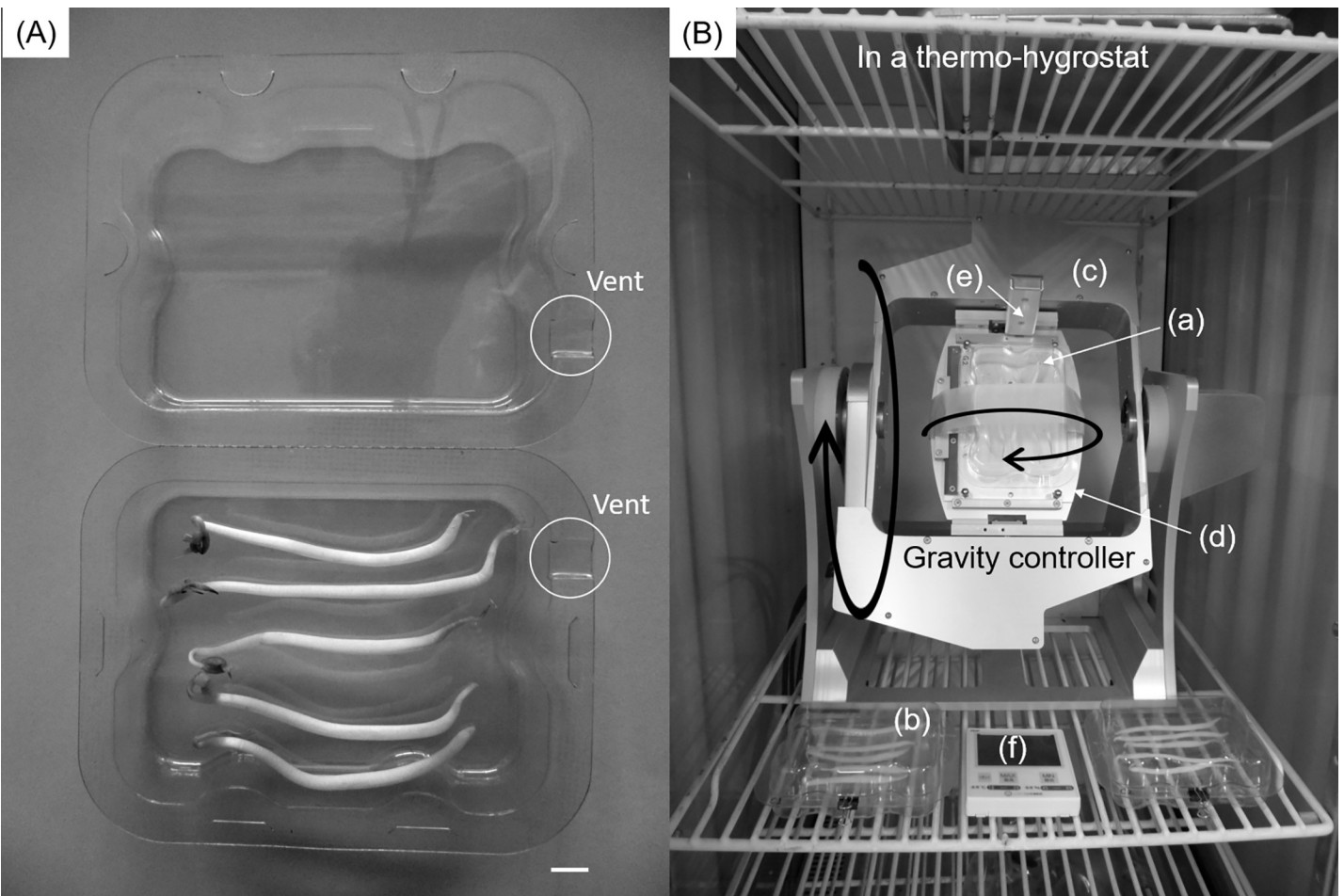

**Fig 1.** (A) A box (A-04 Fix Box, AS ONE Corp., Osaka) to store samples (scale bar: 10 mm) and (B) Method to store vegetables under different gravity levels in a thermohygrostat. (a) Samples (e.g., mung bean sprouts) stored under simulated microgravity in a box set on the inner frame of the gravity controller (Gravite®, Space Bio-Laboratories Co., Ltd. Hiroshima, Japan) (new treatment) (b) samples stored under normal gravity in a box (control treatment) (c) inner flame (d) outer flame (e) accelerometer (f) hygrothermograph.

where $k$ denotes the mass reduction coefficient of a sample ($d^{-1}$), $t$ denotes the storage period (d), and subscript $e$ denotes the assumed equilibrium.

Eq 2 was transformed to a linear model as Eq 3 and mass retention rare values measured in Subsection 2.4 were substituted into the following equation:

$$\ln (M_t - M_e) = \ln (M_0 - M_e) - k \bullet t \tag{3}$$

## Statistical analysis

The mean data for mass retention rates and $k$ values were compared using Student's t-test ($p < 0.05$) with JMP® Pro ver. 14.2.0 software (SAS Institute Inc., Cary, NC, USA). A two-way analysis of variance (ANOVA; $p < 0.05$) and linear regression analysis were additionally performed using the same software.

## Results and discussion

Fig 2 displays the changes over time in the measured values of gravitational acceleration, calculated with the accelerometer installed in the gravity controller. The gravitational acceleration

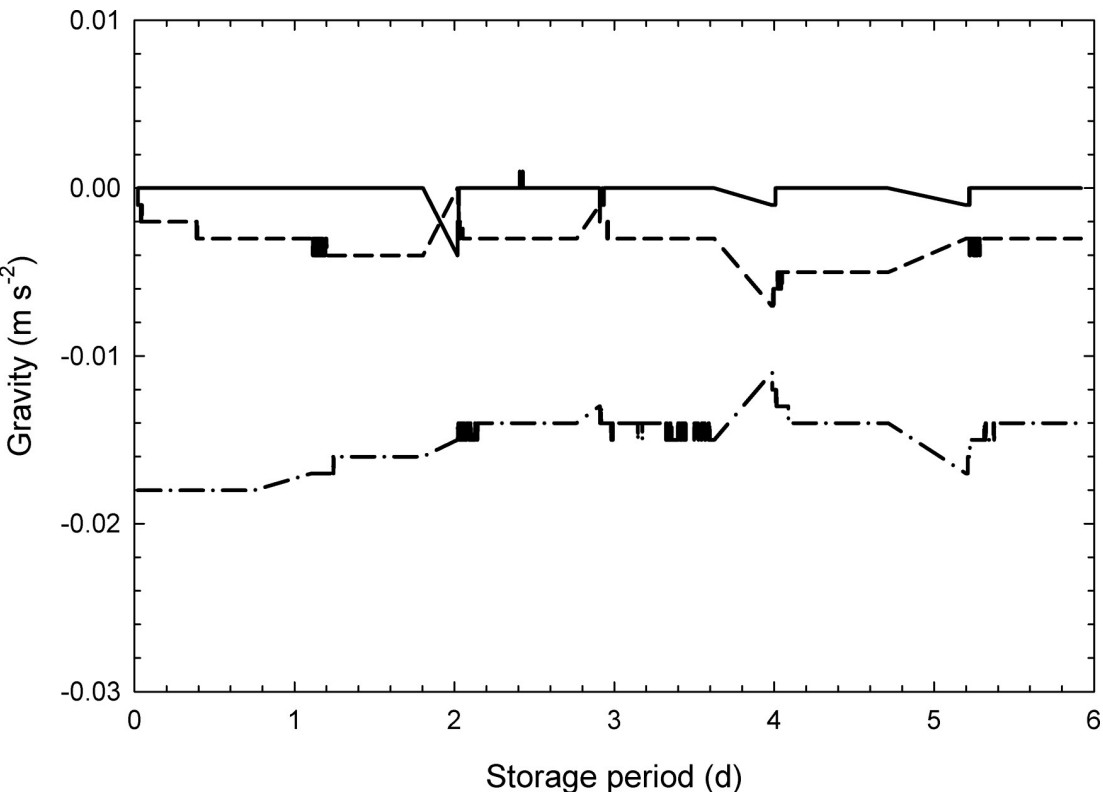

**Fig 2. Changes in gravity on the gravity controller (e.g., mung bean sprouts).** Full line: x axis, dashed line: y axis, chain line: z axis.

under the experimental environment where the vegetable sample prepared as the control was placed measured 9.8 m s$^{-2}$, and the sample attached to the gravity controller was stored in an environment where the gravitational acceleration measured ca. 0 m s$^{-2}$. The principle of operation of the microgravity device is to rotate the object 360° slowly in order to bring the mean value of gravity close to 0 m s$^{-2}$ [26]. Within this study, it was considered possible to conduct an experiment to adequately investigate the effect of gravity on the mass loss of postharvest vegetables.

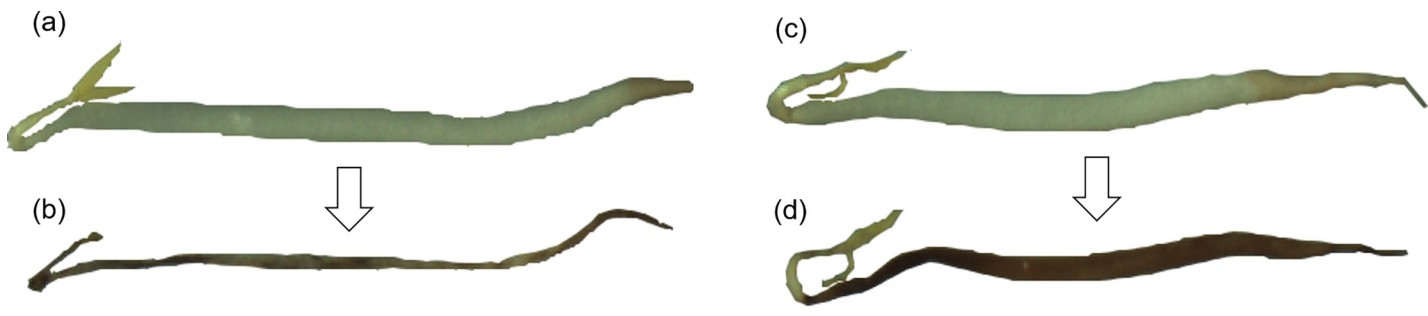

**Fig 3. Typical changes in the shape of mung bean sprouts at different gravity levels (scale bar: 10 mm).** (a) a sample before the start of storage under normal gravity [mass retention rate ($M_r$): 100%], (b) a sample after storage under normal gravity ($M_r$: 5.5%), (c) a sample before the start of storage under simulated microgravity ($M_r$: 100%), (d) a sample after storage under simulated microgravity ($M_r$: 17.8%).

Fig 3 displays the shapes of the sampled mung bean sprouts before storage and after 6 d storage. A comparison between Fig 3(A) and 3(C) before storage demonstrated similar shapes, but the shapes following storage under normal gravity (Fig 3(B)) and simulated microgravity (Fig 3(D)) were varied, respectively. The thickness of the vegetable sample in Fig 3(D) was retained at higher rates than in Fig 3(B). Nagano and Shimaji [27] reported that the higher the levels of the moisture content in vegetables, the wider the plant width, and Bovi et al. [15] found that the mass loss of vegetables postharvest was mainly attributed to moisture loss. According to the aforementioned findings, it was thought that the thickness of mung bean sprouts was retained under the simulated microgravity environment because moisture levels were retained in the experimental conditions.

Fig 4 displays the changes in the mass retention rates of the three types of studied vegetables over time. The principle of operation of the microgravity device is to rotate the object 360° slowly in order to bring the mean value of gravity close to 0 m s$^{-2}$ [26]. Normal gravity was 9.8 m s$^{-2}$ (gravitational acceleration on earth). In the case of vegetable soybean, the difference in gravity levels did not affect the mass retention rate during the storage period for 9 d. In contrast, in the case of mung bean sprouts, the mass retention rate was significantly higher in the simulated microgravity environment, on and after 3 d following the initiation of storage, and in the case of white radish sprouts, on and after 1 d following the beginning of storage. This suggests that the simulated microgravity environment possesses a mass retention effect, and that the effect varies depending on the type of vegetable present under these conditions.

Table 1 displays the results of a two-way ANOVA of the effects of storage gravity environment and storage period on the mass retention of three types of vegetables. The mass retention rate of all three types of vegetable decreased significantly in relation to the storage period, though the effect of gravity was limited to mung bean sprouts and white radish sprouts. This result also supports the result of Fig 4.

Fig 5 displays the results of analysis using Eqs 2 and 3 assuming that the mass reduction is due to moisture loss. In this study, $M_e$ of vegetables was substituted into Eq 3 as vegetable soybean: 28.3% [28], mung bean sprouts: 1.3% (minimum measured data), and white radish sprouts: 1.1% (minimum measured data). According to the result from Fig 5(A), when the measured data of Fig 4 was substituted into Eq 3, a linear relationship was observed. This suggests that the temporal change of the mass retention rate can be analyzed using Eq 2 in the same format as the moisture diffusion rate equation. According to the results in Fig 5(B), the simulated microgravity significantly affected the $k$ values of mung bean sprouts and white radish sprouts, though it did not affect the $k$ values of vegetable soybean. It is possible that the $k$ value was significantly suppressed, and that the mass retention rate was significantly high under the simulated microgravity environment on the basis of the results mentioned above. Since the $k$ values of mung bean sprouts and white radish sprouts were higher than those of vegetable soybean, it was taken into account that the decrease in mass retention rates of sprout vegetables with time were deemed remarkable. It was confirmed that the sprout vegetables demonstrated a significant mass loss compared to the grain studied and that it was difficult for sprout vegetables to maintain their freshness.

According to the results in Fig 4, mass loss of vegetables cannot be suppressed even under the microgravity condition. However, in the present study, simulated microgravity exhibited the effect of preserving the mass retention rate of sprout vegetables such as mung bean sprouts and white radish sprouts. Because vegetable soybean is a product obtained by harvesting a portion of the grain, it is thought that the moisture lost from the surface of the sheath led to subsequent mass loss. This mechanism of mass loss may not be affected dependent of gravity levels. In contrast, the two types of sprout vegetables possess roots, stems and leaves, and it is possible that, in addition to the dissipation of moisture from the plant surface, moisture also moved

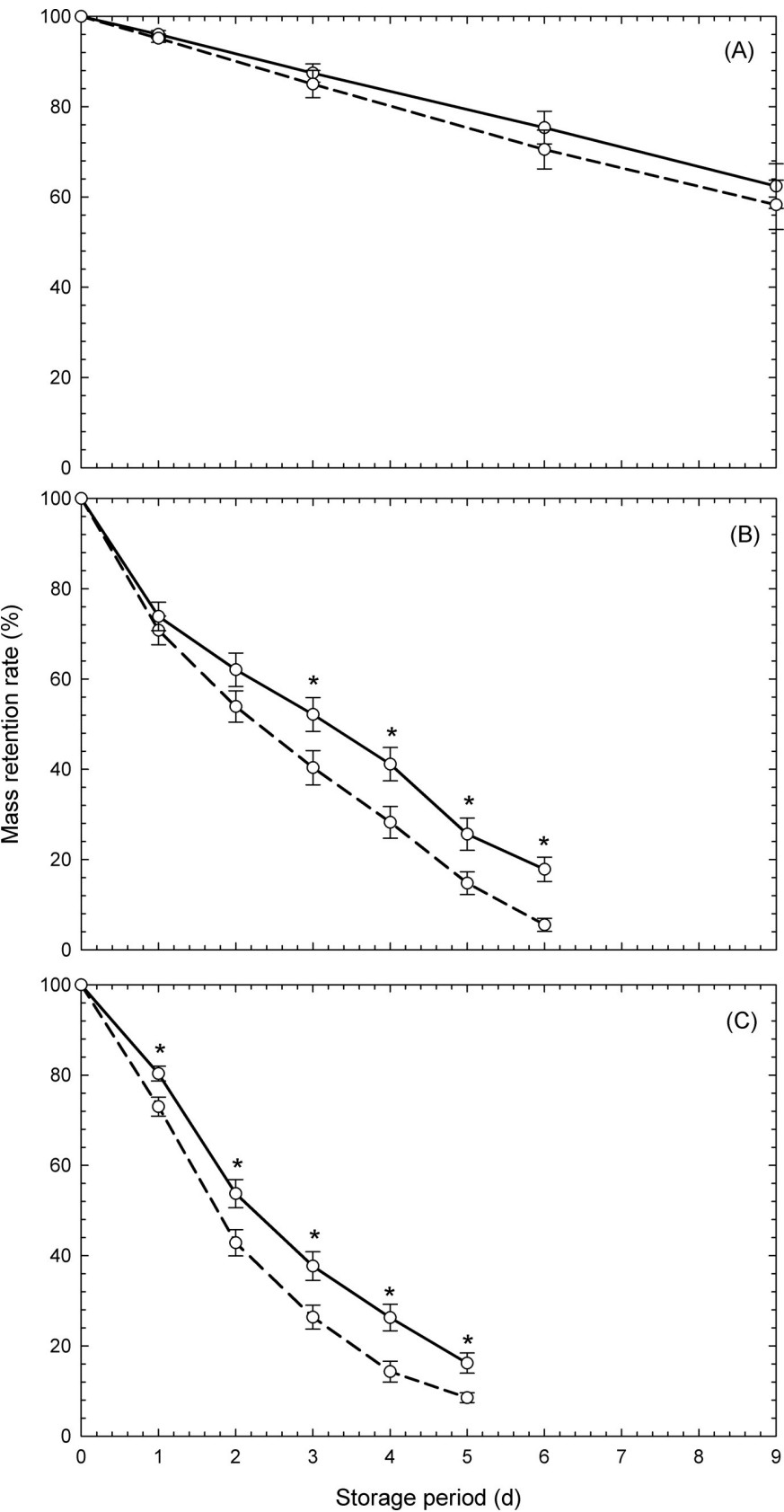

**Fig 4.** Changes in mass retention rates of (A) vegetable soybeans, (B) mung bean sprouts, and (C) white radish sprouts. Symbols are the means of ten (A, B) and six (C) measurements ± standard error. Dashed and full lines denote the data under normal gravity and simulated microgravity levels, respectively. Asterisks denote significant difference at $p < 0.05$ by Student's t-test on the same day and vegetable.

from the roots through the plant canal, leading to transpiration from the leaves. The $k$ value in Fig 5(B) is considered to be the value corresponding to the combined moisture loss by transpiration and moisture loss from the plant surface. Regarding the effect of microgravity on plant tissue, there are several studies utilizing *Arabidopsis thaliana* (L.) Heynh as a sample. Leitz et al. [29] reported that gravitational sensing sites such as endothelial cells and columella cells were deformed and activated when the kinetic energy of statolith sedimentation was converted into biochemical signals. Toyota et al. [30] reported that sedimentation leads to the activation of mechanosensitive $Ca^{2+}$ channels, resulting in elevated $Ca^{2+}$ concentrations in plants. Under the simulated microgravity environment, the direction of gravity is not able to be sensed, the moisture loss due to transpiration is suppressed, and the $k$ value decreases, so it is possible that the mass was retained.

The decline in mass (i.e., mostly moisture) of vegetables lowers nutritional quality, salability (due to wilting, shriveling, softening, increased flaccidity, limpness, loss of crispness, and juiciness), and economic income, due to the loss of salable mass [31]. Therefore, mass retention is important in maintaining freshness, and the mass retention of vegetables by refrigeration [6], CA storage [32], MAP [33], though there are no reports regarding mass retention of vegetables stored in microgravity environments. Although the range of the data in the present study is limited to sprout vegetables, it was suggested by the results that a simulated microgravity environment is effective as a novel freshness preservation technique for postharvest vegetables. When simulated microgravity is employed in combination with conventional storage technologies such as low temperature and modified atmosphere (low $O_2$, high $CO_2$), it is possible that a synergy mass retention effect may be observed.

**Table 1. Two-way analysis of variance on the effect of gravity and storage period on mass retention rate of vegetables.**

| Vegetable | Effect | Degree of freedom | Sum of square | F | p |
|---|---|---|---|---|---|
| Vegetable soybean | Gravity | 1 | 189 | 3.00 | <0.0875 |
| | Storage period | 3 | 14200 | 75.5 | <0.0001*** |
| | Interaction | 3 | 47.8 | 0.253 | <0.859 |
| | Error | 72 | 453 | | |
| | Total | 79 | 19000 | | |
| Mung bean sprout | Gravity | 1 | 2910 | 27.2 | <0.0001*** |
| | Storage period | 5 | 52600 | 98.3 | <0.0001*** |
| | Interaction | 5 | 344 | 0.643 | <0.668 |
| | Error | 108 | 11600 | | |
| | Total | 119 | 67500 | | |
| White radish sprout | Gravity | 1 | 1450 | 38.9 | <0.0001*** |
| | Storage period | 4 | 31300 | 210 | <0.0001*** |
| | Interaction | 4 | 57.0 | 0.381 | 0.821 |
| | Error | 50 | 1870 | | |
| | Total | 59 | 34700 | | |

*** Significant at 99.9% level.

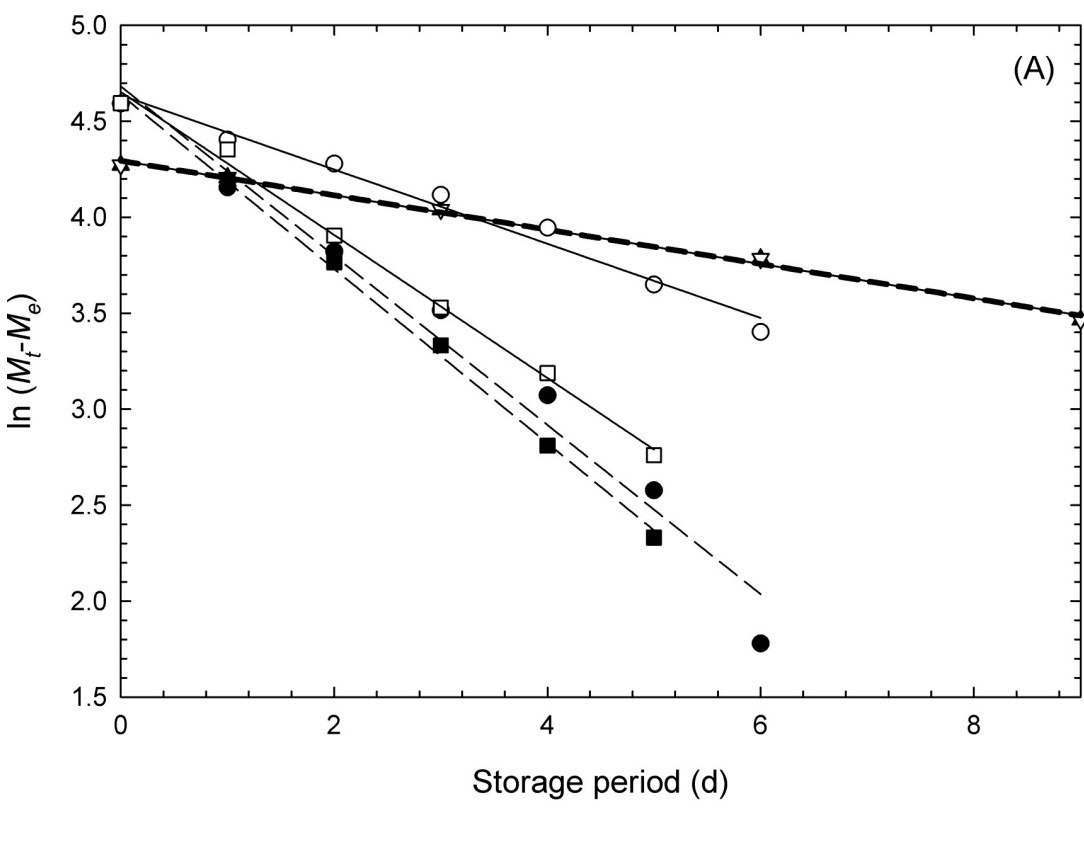

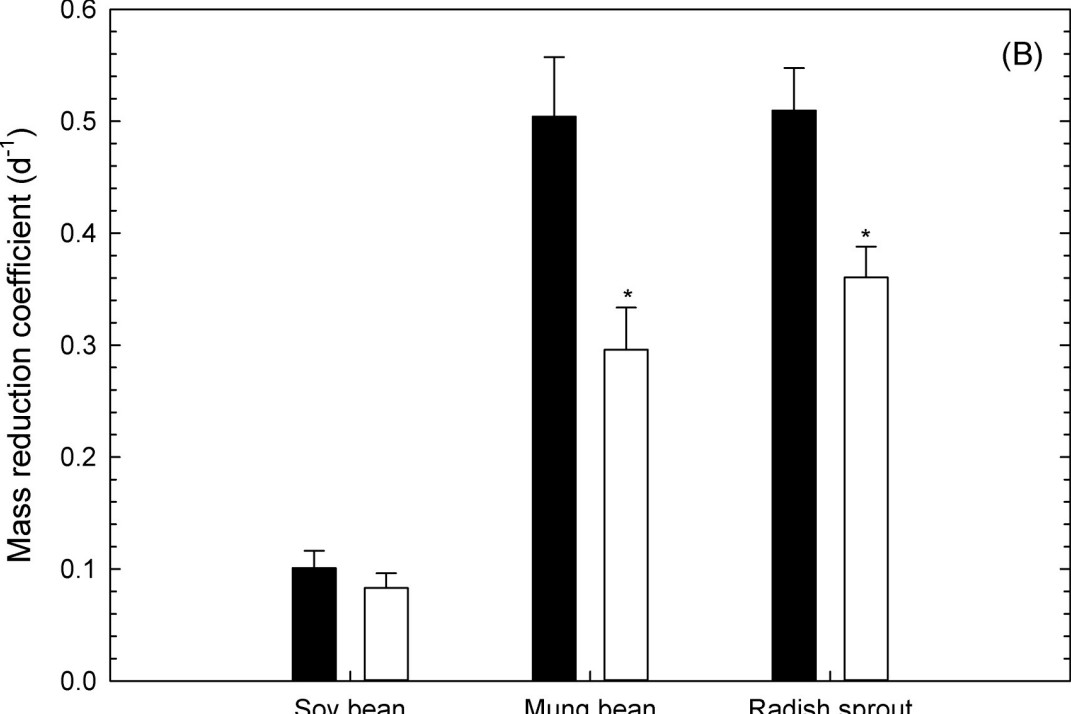

**Fig 5.** (A) Typical linear regression for calculating mass reduction coefficients of vegetable soybeans (triangle), mung bean sprouts (circle), and white radish sprouts (square) stored under normal (closed symbols and dashed lines) and under micro-(open symbols and full lines) gravities; $M_t$ and $M_e$ denote the mass retention rate over the course of storage and assumed

equilibrium mass retention rate, respectively. (B) Means ± ten (vegetable soybeans, mung bean sprouts) and six (white radish sprouts) measurements ± standard error of mass reduction coefficients. Closed and open bars denote the data under normal gravity and simulated microgravity levels. Asterisks denote significant difference at $p < 0.05$ by Student's t-test on the same vegetable.

In addition, data on vegetable freshness in a simulated microgravity environment may be useful for space development research. Several studies on plant cultivation in space exist in current literature. Kitaya et al. [34] cultivated tomato seedlings in a Controlled Ecological Life Support System and promoted growth by supplying $CO_2$. Takahashi et al. [35] investigated the effect of auxin on tropism by performing a cucumber germination experiment aboard a space shuttle. Paul et al. [22] introduced many reports on plant biology enabled by the space shuttle, e.g. Brassica plants on STS-87 had altered chloroplast morphology and altered chlorophyll *a/b* ratios. Since no data have been found concerning changes in the freshness of postharvest vegetables in outer space, it is hoped that the results of this study will be confirmed in outer space in the future.

## Conclusions

The overall effect of preserving the freshness of postharvest vegetables under a simulated microgravity environment produced by slowly rotating an object by 360˚ was investigated in this study. The mass loss of two types of vegetables, mung bean sprouts and white radish sprouts, was suppressed by placing them under a simulated microgravity environment. Since mass loss is a serious phenomenon of freshness degradation among harvested vegetables, the effect of preserving the freshness of vegetables was recognized in the simulated microgravity environment. In contrast, in the case of vegetable soybeans, the suppression of mass loss in the samples was not observed under the pseudo-microgravity environment. According to the aforementioned results, it was confirmed that the simulated microgravity environment possesses the effect of preserving the freshness of the studied sprout vegetables. Since the mass reduction coefficient of the mathematical model for calculating the mass reduction velocity was minimized to be smaller under the simulated microgravity environment than under the normal gravity environment, the freshness preservation effect under the simulated microgravity environment was quantitatively proved. This founding will support long-term space flight missions by prolonging shelf life of sprout vegetables.

## Acknowledgments

We thank D. Ogawa (SBL) for his support on preparation and operation of Gravite®.

## Author Contributions

**Conceptualization:** Yoshio Makino.

**Data curation:** Yumi Kawahara.

**Formal analysis:** Kanji Ichinose.

**Investigation:** Yoshio Makino, Louis Yuge.

**Methodology:** Kanji Ichinose.

**Project administration:** Yoshio Makino.

**Resources:** Yumi Kawahara.

**Software:** Masatoshi Yoshimura.

**Supervision:** Yoshio Makino, Louis Yuge.

**Visualization:** Masatoshi Yoshimura, Louis Yuge.

**Writing – original draft:** Yoshio Makino.

**Writing – review & editing:** Yumi Kawahara.

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
