## [Decision Letter · Decision Letter 0]

10 Sep 2020

PONE-D-20-24552

Depression of mass loss of sprout vegetables under simulated microgravity

PLOS ONE

Dear Dr. Makino,

Thank you for submitting your manuscript to PLOS ONE. After careful consideration, we feel that it has merit but does not fully meet PLOS ONE’s publication criteria as it currently stands. Therefore, we invite you to submit a revised version of the manuscript that addresses the points raised by a reviewer in the field and editorial comments during the review process.

We look forward to receiving your revised manuscript.

Kind regards,

Dr. Sakamuri V. Reddy

Academic Editor

PLOS ONE

Journal Requirements:

2.Thank you for stating the following in the Financial Disclosure:

[The funders had no role in study design, data collection and analysis, decision to publish, or preparation of the manuscript.]. 

We note that one or more of the authors have an affiliation to the commercial funders of this research study : [Space Bio-Laboratories Co., Ltd].

Additional Editor Comments (if provided):

The authors demonstrate that simulated microgravity is effective over conventional methods such as refrigeration to preserve the nutritional value of sprout vegetables. They should consider rephrasing the title, ex., “Efficient preservation of sprouting vegetables under simulated microgravity conditions”. They have observed that vegetable sprouts placed in a simulated microgravity environment preserve the mass maintaining freshness over time. However, they should clarify the implications of the study with respect to space environment or during long-term space flight missions in conclusions of Abstract, Introduction and Conclusions noted (p.11; line 262). Methods-(p.6; line 136) they should clarify the section for statistical significance considered. Also, they should provide microgravity simulation method with citation. Provide a rationale for Fig.2 and Fig.4 under Results and Discussion (p.6; line 142 & p.7; line 171). Fig.3 provided appears to be an illustration. They should show real picture or image of mung bean sprouts under experimental conditions. They should explain the Fig.5B results clearly. The figure shows closed bars and open bars representing normal and microgravity conditions, however no significant change in Soybean. They should explain the results and Figure.5 clearly for the decrease in mass loss under microgravity conditions. I suggest the authors to proofread and take language assistance in preparing the manuscript.

Reviewers' comments:

Reviewer's Responses to Questions

**Comments to the Author**

1. Is the manuscript technically sound, and do the data support the conclusions?

Reviewer #1: Yes

2. Has the statistical analysis been performed appropriately and rigorously? 

Reviewer #1: Yes

3. Have the authors made all data underlying the findings in their manuscript fully available?

Reviewer #1: Yes

4. Is the manuscript presented in an intelligible fashion and written in standard English?

Reviewer #1: Yes

5. Review Comments to the Author

Reviewer #1: This is a clearly written, easy to read research article. It approaches the concept of using a clinostat from a completely new angle - to increase shelf life and improve food preservation of sprouts. While not necessary to add to this current work, I noted the mention of ethylene scrubbing as a means of food preservation, and I am curious if the author has considered the problems that plants often encounter with ethylene in an enclosed environment, specifically on clinostats and in the true microgravity of spaceflight. Ray Wheeler has several publications on this topic, and it seems odd that they were not addressed nor mentioned. I encourage them to look into the topic.

6. PLOS authors have the option to publish the peer review history of their article (what does this mean?). If published, this will include your full peer review and any attached files.

Reviewer #1: **Yes: **C.M. Johnson

---

## [Author Response · Author response to Decision Letter 0]

1 Oct 2020

We have submit "Response to Reviewers.docx" through this submission system to respond to specific reviewer and editor comments.

---

## [Editor Report · Decision Letter 1]

5 Oct 2020

Efficient preservation of sprouting vegetables under simulated microgravity conditions

PONE-D-20-24552R1

Dear Dr. Makino,

We’re pleased to inform you that your manuscript has been judged scientifically suitable for publication and will be formally accepted for publication once it meets all outstanding technical requirements.

Kind regards,

Dr. Sakamuri V. Reddy

Academic Editor

PLOS ONE

---

## [Editor Report · Acceptance letter]

7 Oct 2020

PONE-D-20-24552R1 

Efficient preservation of sprouting vegetables under simulated microgravity conditions 

Dear Dr. Makino:

I'm pleased to inform you that your manuscript has been deemed suitable for publication in PLOS ONE. Congratulations! Your manuscript is now with our production department. 

Kind regards, 

on behalf of

Dr. Sakamuri V. Reddy 

Academic Editor

PLOS ONE